# Adsorption of Ammonium Ions and Phosphates on Natural and Modified Clinoptilolite: Isotherm and Breakthrough Curve Measurements

Kateryna Stepova [1], Iryna Fediv [1], Aušra Mažeikienė [2,*], Julita Šarko [2] and Jonas Mažeika [3]

[1] Department of Environmental Safety, Lviv State University of Life Safety, 35 Kleparivska Str., 79000 Lviv, Ukraine; katyastepova@gmail.com (K.S.); ira.arnaut94@gmail.com (I.F.)

[2] Department of Environmental Protection and Water Engineering, Faculty of Environmental Engineering, Vilnius Gediminas Technical University, Saulėtekio al. 11, LT-10223 Vilnius, Lithuania; julita.sarko@vilniustech.lt

[3] Closed Joint Stock Company "Biotechnologijos Grupė", Beržų Str. 19, Riešės Village, LT-14266 Vilnius, Lithuania

\* Correspondence: ausra.mazeikiene@vilniustech.lt

**Abstract:** The research focuses on ammonia and phosphate removal from wastewater by using a novel metal and microwave-treated clinoptilolite. For increasing adsorption capacity, the samples were calcinated or microwave irradiated in the solutions of Fe(III), Cu (II), or Ca(II) chlorides. BET-specific surface area measurement revealed that the calcination led to a decrease from 18.254 to 11.658 $m^2/g$. The adsorption results were fitted to theoretical models. The $PO_4^{3-}$ adsorption in all samples as well as $NH_4^+$ adsorption in natural and Fe- and Ca-modified samples is best described using the Langmuir–Freundlich model, but in calcinated and Cu-modified clinoptilolite the $NH_4^+$ sorption is better characterized by the Freundlich model. The $PO_4^{3-}$ adsorption in natural and all modified samples is best described using the Langmuir–Freundlich model. Fe-modified and calcinated clinoptilolite showed the highest $NH_4^+$ adsorption capacity of 4.375 and 2.879 mg/g. Ca-modified samples demonstrated the lowest adsorption capacity of 0.875 mg $NH_4^+$/g. The metal-modified samples exhibit a significantly higher phosphate sorption capacity (from 800.62 for Cu-sample to 813.14 mg/g for the Fe-modified sample) than natural (280.86 mg/g) or calcinated samples (713.568 mg/g). Experimental studies in dynamic conditions revealed high $NH_4^+$ and sufficient $PO_4^{3-}$ ions captured on modified clinoptilolite. This study provides a feasible approach for the synchronous removal of the main eutrophication agents for implementation in additional (tertiary) wastewater treatment facilities.

**Keywords:** phosphates; ammonium; clinoptilolite; adsorption; microwave treatment; chemical modification





## 1. Introduction

Wastewater from private households and many industrial and agricultural sectors is characterized by high levels of ammonium and phosphates. However, nitrogen and phosphorus are also two essential nutrients for plants and microorganisms; however, their excess amount may cause an adverse impact on aquatic ecosystems, resulting in eutrophication [1,2]. The presence of nutrients leads to cyanobacteria (blue-green algae) development. Excessive algae growth impairs the operation of water intake facilities and fisheries reduce the hydraulic parameters of the stream, and water blooms also lead to a decrease in dissolved oxygen, deterioration of conditions for the development of flora and fauna, and disruption of the normal functioning of natural ecosystems [3,4]. The high content of phosphate in household wastewater is not only a current problem but also a problem of the last decade. The average phosphate flow is 33.9 thousand tonnes per year. Using the area of Ukraine's territorial waters, it is calculated that the coastal waters of Ukraine with atmospheric precipitation annually receive 1.56 thousand tonnes of phosphorus [4]. The entire territory of Lithuania lies within a zone sensitive

to eutrophication; for this reason, relatively stricter wastewater treatment requirements are applied. Amendments to Wastewater Management Regulation No. D1-236 (2016) of the Republic of Lithuania describes the removal process for phosphorus (up to 5 mg/L) and nitrogen compounds (up to 25 mg/L) in small-scale wastewater treatment plants. The maximum permissible concentration of ammonium nitrogen in treated wastewater discharged into the natural environment is 5 mg/L. Ammonium nitrogen, usually referred to as nitrogen in the form of free ammonia nitrogen ($NH_3$) and ammonium ions ($NH_4^+$), is present in natural waters. Its high content is found in wastewater, such as domestic sewage. The pH of domestic wastewater is usually <9, so the dominant form is $NH_4^+$.

Currently, several valuable technologies for removing nitrogen and phosphorus have been proposed, including biological nutrient removal [5,6], chemical precipitation [5,7], membrane processes, electrolytic treatment, ion exchange, and adsorption [8–10]. After the biological treatment by activated sludge, there is an excessively high content of nitrates and ammonium nitrogen remaining in wastewater [11]. Ammonium nitrogen remains in wastewater after biological treatment when the nitrification process in wastewater treatment plants is disrupted. Ammonium nitrogen is much more toxic than nitrate nitrogen. Several mechanisms have been proposed to explain the toxic effects of ammonium nitrogen: the assimilation process of ammonium nitrogen disrupts the carbon and nitrogen balance in plants, the intracellular pH balance, the balance of nutrients, and leads to energy loss [12,13]. Conventional biological methods may remove only 30–50% of phosphorus from wastewater. In treated wastewater, phosphorus is mostly in the form of phosphate. The solubility of phosphorus in sewage sludge can be increased by ozonation [14], but this must be followed by phosphate retention in the adsorption column. For low concentrations of phosphate ions, biotreatment and precipitation methods are not effective [8]. Existing biological methods are not able to achieve the required treatment level from phosphorus and nitrogen compounds, and physicochemical methods, while showing fairly good treatment results, require significant costs and, furthermore, cause the need to process sludge generated during reactive treatment [15].

Adsorption is probably the key to the effective removal of N and P from the environment [16]. Multiple materials, including zeolites [17,18], activated carbon [19–21], metal oxides [22,23], etc., have recently been investigated as nitrogen and phosphorus adsorbents. However, there is a particular interest in adsorbents that can efficiently treat low-concentrated solutions and ensure low residual content in the effluents. The requirements of the legislation of the European Union and other countries set high standards for municipal wastewater treatment. Zeolites are considered to be promising materials for water treatment due to their high removal efficiency, availability, and affordability. Clinoptilolite, being one of the most common natural zeolites, has a number of valuable physicochemical properties such as cation exchange, catalysis, and molecular filtration [24]. It was suggested as an excellent adsorbent for ammonium removal [25–27]. In order to change the surface charge of natural minerals from negative to positive and to enable them to adsorb phosphate anions, modification is needed [28]. The synthesis of porous materials is very attractive due to their various applications in water treatment [29,30]. Recently, it is observed that nitrates and phosphates can be adsorbed simultaneously by zeolite, supported by Fe/Ni bimetallic nanoparticles [31]. In addition, bifunctional nanocomposite containing implanted Fe(III) hydroxide nanoparticles was tested and found suitable for the simultaneous removal of nitrate and phosphate [32]. Although, there is a paucity of scientific literature on sorbents that simultaneously remove ammonium nitrogen and phosphate phosphorus from real wastewater. The adsorption capacity of natural minerals for phosphate is significantly improved by modification with polyvalent metals such as $Zr^{4+}$ [33], $La^{3+}$ [34], $Al^{3+}$ [35], or $Fe^{3+}$ [33], but in recent decades, a promising way for improving the efficiency of sorbents is activating them using microwave radiation. Most of the studies in this area [36–38] show the samples of individual synthetic sorbents irradiated with microwaves under different conditions. Under the influence of microwave radiation, new microcracks appear in them and large grains are crushed, which significantly increases

the active surface area of the sorbent; moreover, irradiation of sorbents directly in salt solutions significantly increases the amount of metal precipitated on the mineral's surface. The present research focuses on ammonia and phosphate removal from aqueous solutions by using a novel metal and microwave-pretreated clinoptilolite. A new modification method of clinoptilolite was proposed, samples modified in four ways were produced and tested, and their properties were compared with the properties of natural zeolite. The authors hypothesized that clinoptilolite modified by microwaves and metals will be suitable for the simultaneous removal of ammonium nitrogen and phosphates from wastewater. The aim was to find the most efficient modification for the simultaneous removal of coexisting contaminants from water. The study results will be useful in creating additional (tertiary) wastewater treatment facilities.

## 2. Materials and Methods

### 2.1. Characterization of Adsorbents

Natural clinoptilolite (pH of aqueous extract–7.75; bulk density–947 kg/m$^3$) was taken for sorption studies. Clinoptilolite is a natural type of zeolite with a cage-like structure and high cation exchange capacity; therefore, it is suitable for the capture of ammonium ions. In order to sorb negatively charged phosphate ions, the clinoptilolite had to be modified by adding Fe, Ca, and Cu ions. Nanoscale iron or iron oxides attract negatively charged nitrate and phosphate ions [39]. Calcination increases the phosphorus adsorption capacity of sorbents [40]. Copper is one of the most widespread metals; it is essential to all living organisms, has low toxicity, and is cheaper than other metals [41]. Copper oxide has the potential for being an effective sorbent of phosphate under optimum conditions [42]. Samples for synthesis were prewashed, levigated, and dried at 80 °C until constant weight. After drying, the samples were sieved. A particle size fraction of 0.8–1.2 mm was chosen for research.

For increasing the adsorption capacity of samples, the following types of preliminary treatment were used: calcination at 550 °C for 3 h, and microwave treatment for 10 min at 790 W in the following solutions: 0.12 mol/L FeCl$_3$, 0.15 mol/L CuCl$_2$, and 0.18 mol/L CaCl$_2$. The solutions were prepared from anhydrous salts purchased at Sfera Sim Ltd., Lviv, Ukraine.

The following natural and modified sorbents were chosen for sorption studies:

(1)  CL_nat-natural clinoptilolite;
(2)  CL_thermo-clinoptilolite calcinated;
(3)  CL_Fe-Fe-modified microwaved clinoptilolite;
(4)  CL_Cu-Cu-modified microwaved clinoptilolite;
(5)  CL_Ca-Ca-modified microwaved clinoptilolite.

BET Surface Area Pore Size Distribution

The instrument for recording nitrogen adsorption/desorption isotherms—Quantachrome Autosorb-iQ-KR/MP automated, high-vacuum, gas sorption analyser—was used to determine the specific surface area, pore volume, and pore size distribution of the studied materials. Samples were outgassed under a vacuum at 150 °C for 3 h. Nitrogen adsorption/desorption isotherms were measured at −196 °C (77 K). The BET (Brunauer–Emmett–Teller) equation was used to calculate the specific surface area. The density functional theory (DFT) and QSDFT method were applied to determine the pore size distribution. The total pore volume was measured from the adsorption isotherm using the uptake of nitrogen at a relative pressure of p/p$_0$ = 0.99. All calculations were performed using ASiQwin program (Version 2.0), developed by Quantachrome Instrument, Boynton Beac, FL, USA.

### 2.2. Characterization of Solutions and Wastewater Used for Adsorption

2.2.1. Model Solutions (Used in Static Conditions)

The model potassium phosphate solution was prepared from anhydrous $KH_2PO_4$ (1.9175 g) dissolved in a 1 L flask with 10 mL of $H_2SO_4$ (1.34 g/L) added and made up to the mark with distilled water, then mixed thoroughly.

The model ammonium chloride solution was prepared by dissolving of 0.3 g of $NH_4Cl$ in a 1 L flask and made up to the mark with distilled water, then thoroughly mixed.

A set of working solutions was prepared from the initial solution by repeated dilution with distilled water. Into 100 mL flasks 10, 20, 40, 50, 60, and 80 mL of the model solution were added and made up to the mark with distilled water.

2.2.2. Real Solutions (Used in Dynamic Conditions)

The content of the wastewater used for the study is presented in Table 1.

**Table 1.** Wastewater content after biological treatment.

| $NH_4$-N (mg/L) | $PO_4$-P (mg/L) | $NO_3$-N (mg/L) | BOD (mg/L) | TSS (mg/L) | pH | T (°C) |
|---|---|---|---|---|---|---|
| $9.7 \pm 0.4$ | $3.37 \pm 0.3$ | $1.96 \pm 0.2$ | $5.5 \pm 0.4$ | $6.6 \pm 0.4$ | $7.35 \pm 0.2$ | $20 \pm 0.5$ |

After biological treatment in an individual WTP, small concentrations of organic matter and suspended solids remained in the wastewater: BOD < 6 mg/L; TSS < 7 mg/L. Low concentrations of organics and suspended solids allowed filtration studies to be carried out because the filter fillers did not clog.

### 2.3. Experiments Methodology

2.3.1. Adsorption Isotherms

The sorption properties of the samples were investigated under static conditions.

Ten beakers were filled with 100 mL of working solution of appropriate concentration and 1.0 g of the sample was added, stirred, and left for 24 h. The solutions were then filtered and analysed for the content of $NH_4^+$ and $PO_4^{3-}$ ions.

The content of $NH_4^+$ ions in the solutions was determined by direct potentiometry (Ionometer AI-125, Spectro Lab, Kyiv, Ukraine). The content of $PO_4^{3-}$ ions in the solutions was determined by a photoelectric colorimeter (KFK-2, Zagorsk Optical-Mechanical Plant, Russia). All experiments were repeated three times.

Removal parameters and maximum adsorption capacity at equilibrium were determined by the correlation between the amount of adsorbed ammonium or phosphates $q_e$ [mg/g] and the equilibrium concentration $C_e$ [mg/L]. The adsorption isotherms are described by the following mathematical equations [43].

In Equation (1), $K_L$ is the Langmuir isotherm constant, which characterizes the affinity of the adsorbent and adsorbate, $dm^3/mg$; $q_m$ is the maximum sorption capacity, mg/g of sorbent; $C_e$ and $q_e$ are the equilibrium concentrations of the component in the liquid and solid phases, respectively.

$$q_e = \frac{q_m K_L C_e}{1 + K_L C_e} \tag{1}$$

The Freundlich model (2) is an exponential equation; this equation allows an infinite adsorption process.

$$q_e = K_F C_e^{1 \backslash n_f} \tag{2}$$

$K_F$ is the Freundlich isotherm constant, which characterizes the adsorption capacity, mg/g of sorbent. When the value of $K_F$ increases, the adsorption capacity grows. $n_f$ is the heterogeneity coefficient; therefore, the Freundlich isotherm model can be used for heterogeneous systems.

The Langmuir–Freundlich isotherm model (3) at low concentrations of the adsorbate is reduced to the Freundlich isotherm, while at high concentrations it predicts the adsorption capacity of the monolayer, which is inherent to the Langmuir isotherm.

$$q_e = \frac{q_m(K_{LF}C_e)^{n_{LF}}}{1 + (K_{LF}C_e)^{n_{LF}}} \tag{3}$$

where $q_m$ and $K_{LF}$ are the adsorption capacity and the affinity constant, respectively, and $n_{LF}$ is the coefficient of heterogeneity or a measure of the adsorption intensity. If $n_{LF} = 1$, then Equation (3) reduces to the Langmuir isotherm model.

As the linear modelling is unsuitable for three-parameter isotherms, five different error functions were investigated, and in each case, the isotherm parameters were determined by minimizing the corresponding error function in the range of liquid phase concentrations using the "Solver" add in in the MS Excel spreadsheet. A detailed description of the procedure is presented in [44].

### 2.3.2. Breakthrough Curves
Description of the Experimental Unit

The laboratory stand (Figure 1) was used for discharge equalization and the composition of the wastewater fed to the filter media [45]. The tested zeolite fillers were placed in the filtration columns in threes, then the experiment was repeated with the other two fillers. The height of the columns was 60 cm and the height of sorbent fillers in the tests was 21–21.5 cm. A pump ((3), Figure 1) was applied to supply three filters with transported wastewater at a rate of 0.68 m/h (discharge of 0.96 L/h). Filtrate samples from ((5), Figure 1) were taken every 30 min to measure the pH and concentrations of $PO_4$-P and $NH_4$-N. The study was repeated two more times to present the mean results of three experiments.

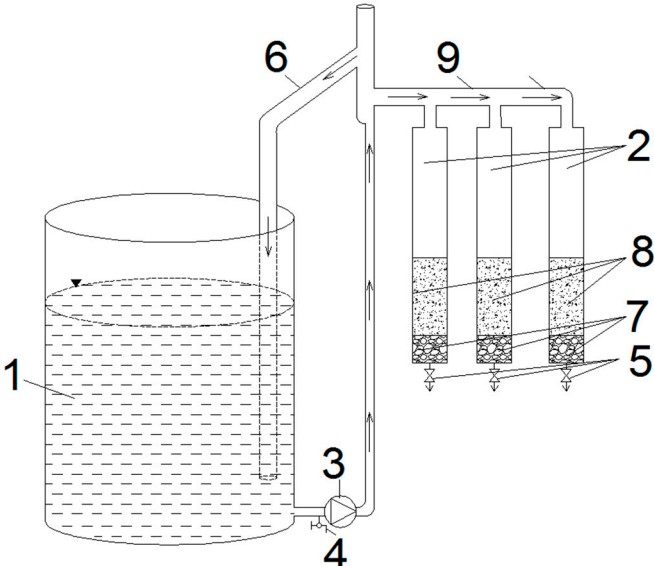

**Figure 1.** Scheme of the experimental stand: 1—wastewater tank, 2—filtration columns, 3—supply pump, 4—sampling point (before filtration), 5—filtrate collection point, 6—overflow tube, 7—retaining layer, 8—filter media, and 9—wastewater distribution pipe to columns.

$PO_4$-P and $NH_4$-N concentrations in the filtrates were determined using MERCK Spectroquant® (Merck KGaA, Darmstadt, Germany) tests, pouring test samples into cuvettes (Hellma GmbH & Co. KG, Müllheim, Germany), and measuring with a Genesys 10 UV-Vis spectrophotometer (Thermo Fisher Scientific Inc., Waltham, MA, USA). The pH of the produced solution was determined by measuring it with an Oxi 330/SET device (WTW

GmbH & Co. KG, Weilheim, Germany). The effectiveness of removing nutrients $E_i$ (%) from wastewater [46] was calculated according to Formula (4):

$$E_i = \frac{C_0 - C_i}{C_0} \qquad (4)$$

where: $C_0$—PO$_4$-P or NH$_4$-N concentration before treatment, (mg/L); $C_i$—PO$_4$-P or NH$_4$-N concentration after treatment, (mg/L).

All measurements were carried out in triplicate. Data were statistically processed using STATGRAPHICS (2018). One-way ANOVA (at a significance level of $p < 0.05$) followed by a Tukey post-hoc test was used to differentiate between the means of the samples.

## 3. Results and Discussion

### 3.1. Surface Area and Porosity

The main textural characteristics of the samples including BET surface area ($S_{BET}$), micropore area ($S_{micro}$), external area ($S_{ext}$), pore volume ($V_{pore}$), and average particle size ($d_p$), along with porosity are displayed in Table 2.

**Table 2.** Specific surface area ($S_{BET}$), micropore area ($S_{mic}$), external surface area ($S_{ext}$), pore volume ($V_p$), equivalent particle size ($d_{part}$), and porosity ($\varepsilon$) of natural and modified clinoptilolite.

| Sample Code | $S_{BET}$, m$^2$/g | $S_{mic}$ | $S_{ext}$ | $V_p$, mL/g | $d_{part}$ | $\varepsilon$ |
|---|---|---|---|---|---|---|
| CL_nat | 18.254 | 4.773 | 13.481 | 0.041 | 347.2 | 0.037 |
| CL_thermo | 11.658 | 0.379 | 11.279 | 0.035 | 543.6 | 0.031 |
| CL_Fe | 19.025 | 7.18 | 11.845 | 0.039 | 333.1 | 0.035 |
| CL_Cu | 15.9 | 5.21 | 10.69 | 0.035 | 398.6 | 0.031 |
| CL_Ca | 14.129 | 3.85 | 10.279 | 0.034 | 448.6 | 0.031 |

The N$_2$ adsorption–desorption isotherms for test materials are presented in Figure 2. Due to the presence of hysteresis loops, all the isotherms can be classified as type IV (IUPAC) [47]. The hysteresis loop is associated with capillary condensation taking place in mesopores, and the limiting uptake over a range of high P/P$_0$. Type IV isotherms characterize a mesoporous material with a high tendency for adsorption. The samples show the loop of H3 type. The shape of the hysteresis loop is often identified with a specific pore structure. The loop of H3 type is observed with aggregates of plate-like particles with slit-shaped pores.

As is seen from Table 2, the calcination at 550 °C leads to the decrease of the BET surface area from 18.254 to 11.658 m$^2$/g in clinoptilolite. Furthermore, the micropore area sufficiently decreases after calcination, as shown in Figure 3, where the pore size distribution is presented. Instead, microwave irradiation of samples does not lead to significant changes in the BET surface area. From these results, we can conclude that calcination causes shrinkage of the pores, as opposed to microwave irradiation which leaves behind numerous open pores. Moreover, pretreatment with FeCl$_3$ and CuCl$_2$ under microwave irradiation leads to an increase in the micropore area; although, the BET surface area slightly decreases. The order for S$_{BET}$ values for clinoptilolite was CL-Fe > CL > CL-Cu > CL-Ca > CL-thermo.

The equivalent particle size ($d_{part}$) was estimated assuming the spherical shape of the particles and using the relation (5) [48]:

$$d_{part} = \frac{6000}{S_{BET} \cdot \rho} \qquad (5)$$

where $\rho$ is the density of bulk composite in g/cm$^3$ and $S_{BET}$ is expressed in m$^2$/g. The obtained results are listed in Table 2. It is clear that the decrease in the BET surface area after calcination is caused by particle agglomeration due to sintering.

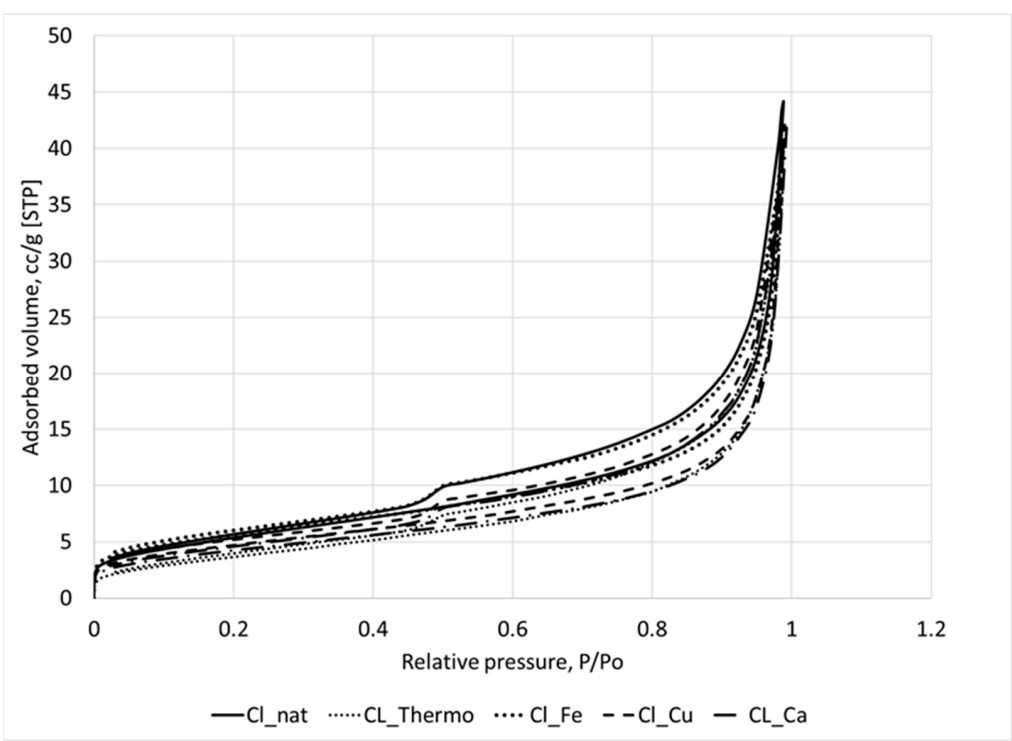

**Figure 2.** N$_2$ adsorption–desorption isotherms.

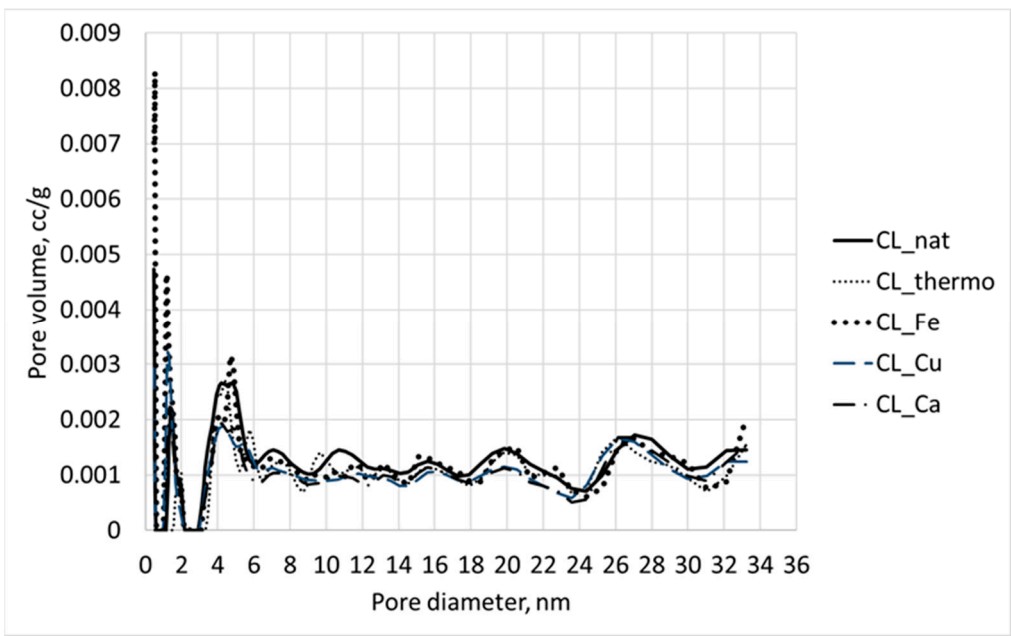

**Figure 3.** Pore size distribution.

The porosity of the particles ($\varepsilon$) is calculated using the following Equation (6) [49]:

$$\varepsilon = \frac{V_p}{V_p + \frac{1}{\rho_{app}}} \tag{6}$$

where $V_p$ is the pore volume (cm$^3$/g) and $\rho_{app}$ is the apparent density (g/cm$^3$) of investigated materials. Porosity is a measure of the void spaces in a material and is a fraction of the volume of voids over the total volume, between 0 and 1. Although the porosity increases

as a result of microwave irradiation, it decreases when irradiated in metal-containing solutions due to the pores filling with Fe or Cu species, as shown in Table 2.

### 3.2. Adsorption Isotherms

The $NH_4^+$ adsorption isotherms on natural and modified clinoptilolite are presented in Figure 4. It is observed that the $NH_4^+$ amount adsorbed varies within a wide range.

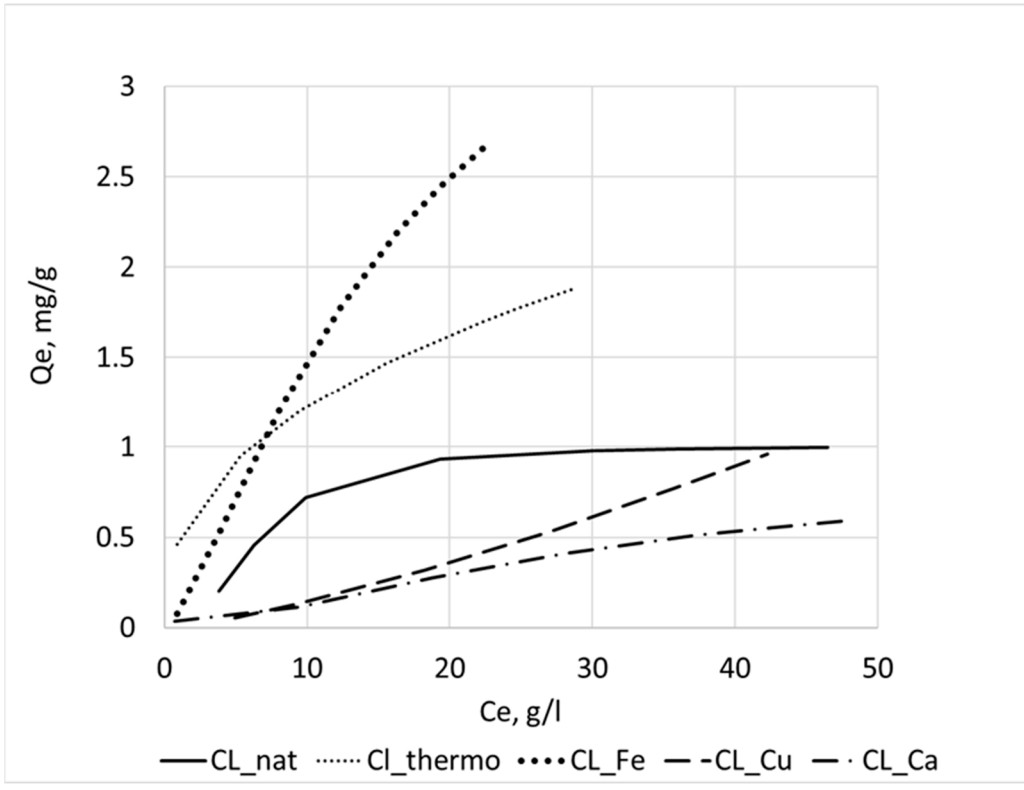

**Figure 4.** Isotherms of $NH_4^+$ adsorption.

The shape of the curve representing the variations of $NH_4^+$ amount adsorbed as a function of pressure corresponds to an IV-type isotherm according to IUPAC classification [47]. An IV-type isotherm is typically characteristic of a mesoporous adsorbent. The results of the nonlinear fitting of experimental research within theoretical models are presented in Table 3.

As evidenced in Table 3, the process of $NH_4^+$ adsorption in natural and Fe- and Ca-modified samples is best described using the Langmuir–Freundlich model, but the process on calcinated and Cu-modified clinoptilolite is better characterized by the Freundlich model. This fact may be evidence of the heterogeneous surface of the adsorbent. Remarkable is the fact that the isotherms of Cu and Ca samples are of type V (mesoporous with weak interaction) and three other samples are of type IV (mesoporous), according to IUPAC [47]. This observation coincides with the data presented in Table 1. Cu- and Ca-modified samples have the smallest external surface area, which may lead to weaker interaction. There is a clear difference between adsorbents since Fe-modified and calcinated clinoptilolite showed the highest $NH_4^+$ adsorption capacity of 4.375 and 2.879 mg/g according to the Langmuir–Freundlich model. Ca-modified samples demonstrated the lowest maximum adsorption capacity of 0.875 mg/g, which is even lower than in untreated natural material (Langmuir–Freundlich model). This fact correlates with their external surface area presented in Table 2, proving the fact that in these materials mesopores are the main adsorption sites. However, the untreated clinoptilolite has the largest specific surface area, but the maximum sorption capacity is not the highest. This observation could be attributed to different kinds of adsorption mechanisms. Most probably, the reason is the additional adsorption centres

created during thermochemical pretreatment. Therefore, the cation exchange mechanism takes place along with the physical adsorption.

**Table 3.** $NH_4^+$ adsorption isotherms nonlinear fitting parameters.

| | Sample Index | | | | |
|---|---|---|---|---|---|
| | **CL_nat** | **CL_thermo** | **CL_Fe** | **CL_Cu** | **CL_Ca** |
| Langmuir isotherm parameters | | | | | |
| $q_m$ | 1.311 | 4.398 | 0.027 | 83.913 | 7.286 |
| $K_L$ | 0.093 | 0.029 | 0.467 | 0.0002 | 0.002 |
| SNE | 3.942 | 3.642 | 3.591 | 3.900 | 3.900 |
| $R^2$ | 0.95 | 0.99 | 0.47 | 0.81 | 0.98 |
| Freundlich isotherm parameters | | | | | |
| $n$ | 3.158 | 2.456 | 1.197 | 0.757 | 3.962 |
| $K_F$ | 0.321 | 0.479 | 0.205 | 0.007 | 0.217 |
| SNE | 3.917 | **3.465** | 3.597 | **3.878** | 3.781 |
| $R^2$ | 0.72 | 1.0 | 0.98 | 1.0 | 0.44 |
| Langmuir–Freundlich isotherm parameters | | | | | |
| $q_m$ | 1.006 | 2.879 | 4.375 | 1.750 | 0.875 |
| $K_{FL}$ | 0.147 | 0.061 | 0.061 | 0.025 | 0.033 |
| $n_{FL}$ | 2.455 | 1.514 | 1.384 | 1.893 | 1.629 |
| SNE | **3.883** | 3.593 | **3.560** | 4.146 | **3.015** |
| $R^2$ | 1.0 | 0.76 | 1.0 | 0.96 | 1.0 |

Values in bold type indicate the isotherm model with minimum SNE value for each sample.

In Figure 5 the $PO_4^{3-}$ adsorption isotherms on natural and modified clinoptilolite are presented.

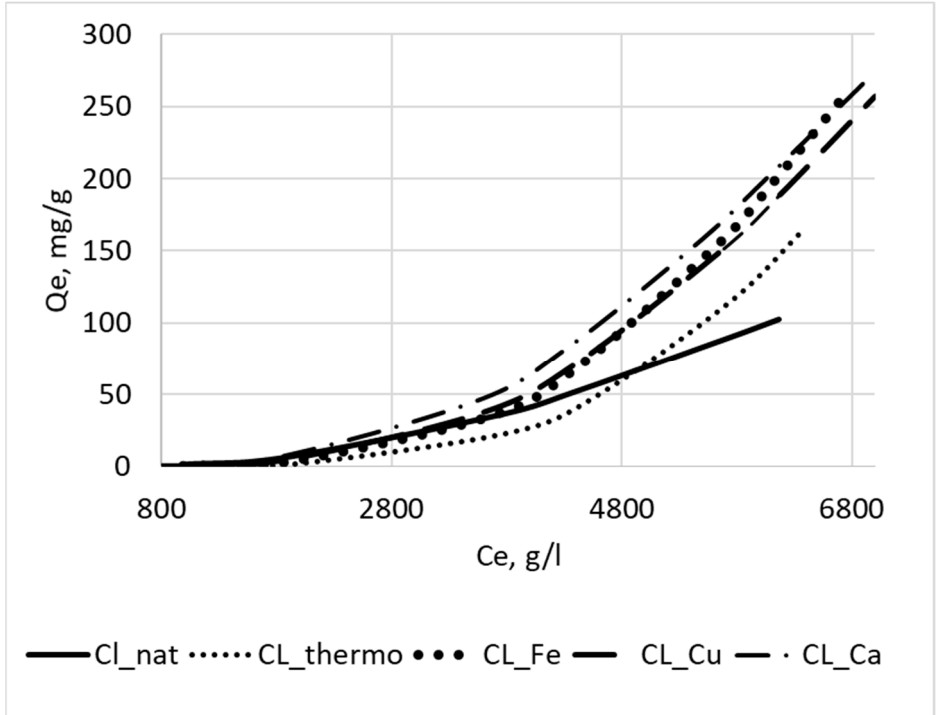

**Figure 5.** Isotherms of $PO_4^{3-}$ adsorption.

It is noteworthy that the amount of adsorbed substance and the types of isotherms differ slightly. All the isotherms are of type V (mesoporous with weak interaction) [47]. The difference between the shapes of the phosphate adsorption curves and the ammonium

adsorption isotherms is explained by a different mechanism of the process. Clinoptilolite behaves as a sorbent for $PO_4^{3-}$ and as an ion exchanger for $NH_4^+$ ions [50].

The results of the nonlinear fitting of experimental research within theoretical models are presented in Table 4.

**Table 4.** $PO_4^{3-}$ adsorption isotherms nonlinear fitting parameters.

| | Sample Index | | | | |
| --- | --- | --- | --- | --- | --- |
| | **CL_nat** | **CL_thermo** | **CL_Fe** | **CL_Cu** | **CL_Ca** |
| | Langmuir isotherm parameters | | | | |
| $q_m$ | 200.0 | 1127.41 | 700.00 | 1989.83 | 2745.80 |
| $K_L$ | $4.21 \times 10^{-5}$ | $6.30 \times 10^{-6}$ | $1.58 \times 10^{-5}$ | $7.63 \times 10^{-6}$ | $1.21 \times 10^{-5}$ |
| SNE | 3.510 | 3.490 | 3.562 | 3.557 | 3.487 |
| $R^2$ | 0.28 | 0.26 | 0.27 | 0.27 | 0.75 |
| | Freundlich isotherm parameters | | | | |
| $n$ | 0.511 | 0.503 | 0.513 | 0.525 | 0.532 |
| $K_F$ | $4.73 \times 10^{-6}$ | $1.41 \times 10^{-6}$ | $8.69 \times 10^{-6}$ | $1.14 \times 10^{-5}$ | $1.49 \times 10^{-5}$ |
| SNE | 3.487 | 3.52 | 3.570 | 3.519 | 3.506 |
| $R^2$ | 0.70 | 0.33 | 0.82 | 0.88 | 0.89 |
| | Langmuir–Freundlich isotherm parameters | | | | |
| $q_m$ | 280.86 | 713.568 | 813.14 | 800.62 | 708.33 |
| $K_{FL}$ | $1.8 \times 10^{-4}$ | $1.18 \times 10^{-4}$ | $0.12 \times 10^{-3}$ | $0.12 \times 10^{-3}$ | $0.12 \times 10^{-3}$ |
| $n_{FL}$ | 5.704 | 4.270 | 3.835 | 3.40 | 3.254 |
| SNE | **3.465** | **3.476** | **3.559** | **3.516** | **3.947** |
| $R^2$ | 0.99 | 1.0 | 0.96 | 0.98 | 0.98 |

Values in bold type indicate the isotherm model with minimum SNE value for each sample.

The process of $PO_4^{3-}$ adsorption in natural and all modified samples is best described using the Langmuir–Freundlich model. This confirms the assumption of phosphate physical adsorption on the heterogeneous surface of the clinoptilolite. Here, the difference in microporosity and the specific surface area does not affect the process efficiency, unlike in the case of ammonium sorption. Nevertheless, the metal-modified samples exhibit a significantly higher phosphate sorption capacity than natural or heat-treated samples. It can be attributed to possible chemical interactions between metal ions and phosphates, causing the formation of insoluble compounds on the surface and in the pores of the adsorbent. According to the authors [51], the sorption mechanisms proved the ion exchange in the case of ammonium and the formation of inner-sphere complexes with the functional groups M-OH (M: $Al^{3+}$, $Fe^{3+}$, $Mn^{2+}$) in the case of phosphate. In this case, Fe- and Cu-modified microwave-irradiated clinoptilolite has the best phosphate sorption characteristics. According to the $q_m$ index, the adsorption capacity of the studied samples creates the following row: Fe- > Ca- > Cu-clinoptilolite, while thermal treatment has almost no effect on the sorption capacity of the sample.

### 3.3. Breakthrough Capacity

Experimental studies were carried out using the described pilot column to establish the dynamic behaviour of $NH_4^+$ and $PO_4^{3-}$ ions captured on natural and modified clinoptilolite.

Figure 6a shows $NH_4^+$ adsorption breakthrough curves on clinoptilolite samples at a concentration of 10 mg/L and a wastewater flow rate of 0.96 L/h at 20 °C.

In Figure 6, natural and calcinated clinoptilolite breakthrough curves show more rapid saturation than all metal-modified samples. Even after 4 h, these samples showed a rather high efficiency of more than 90% (Figure 6). The treatment efficiency of Fe-, Cu-, and Ca-modified samples varies from 99.1 to 99.7%

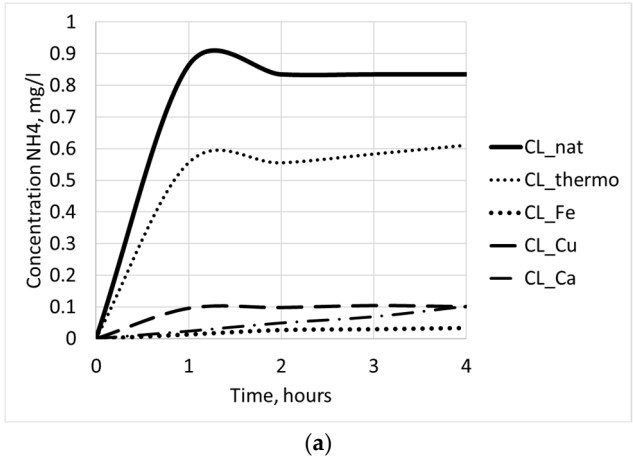 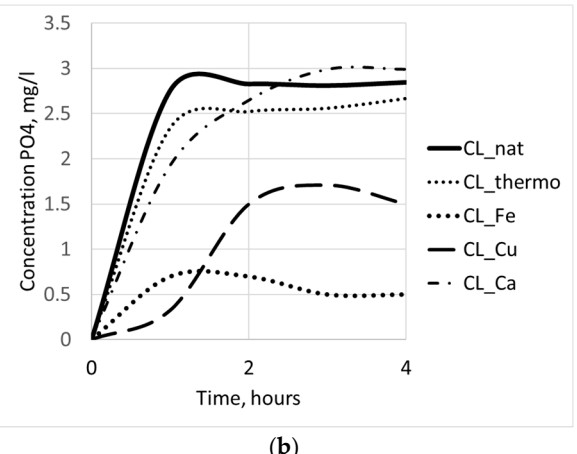

(a)                                   (b)

**Figure 6.** Breakthrough curves: (**a**)—$NH_4^+$; (**b**)—$PO_4^{3-}$.

The $PO_4^{3-}$ breakthrough curve, shown in Figure 6b, reveals quite different dynamic behaviour of the studied samples. Three samples—natural, calcinated, and Ca-modified—reached the saturation point in 1 h with lower efficiency of 16.4%, 21.6%, and 11.2%, respectively (Figure 7). Whereas Fe- and Cu-clinoptilolite didn't reach the saturation even after 4 h, exhibiting the efficiency of 85.6% and 62.6%, respectively (Figure 7). Compared to other sorbents that simultaneously remove nitrogen and phosphorus from wastewater, competitive results were obtained. For example, the commercial sorbent bifunctional nanocomposite HFO@TPR removed nitrate nitrogen and phosphorus from a binary solution with the capacity of 29 mgN/g and 12 mgP/g [32]. The modified palygorskite–bentonite clay removed phosphates and ammonium from water with the capacity of 1.74 mgP/g and 12.87 mgN/g [52]. The potassium clinoptilolite impregnated hydrated metal oxides had a higher sorption capacity; that is 6.8 mg/g of phosphate and 29.0 mg/g of ammonium [51]. The authors [51] note that clinoptilolite impregnated hydrated metal oxides, having absorbed N and P, can be successfully used as a fertilizer. In their study, the sequential fractioning of the loaded modified zeolite revealed the existence of an important fraction of biologically active phosphorus, and—what is more—the recovered phosphate is suitable for fertilizing P-deficient soils. Hydrated metal oxides, such as Fe(III) and Cu(II), have been more extensively explored for phosphate removal [8] because they exhibit strong ligand sorption (of $HPO_4^{2-}$ and $H_2PO_4^-$) through the formation of inner-sphere complexes or through outer-sphere complexes [53]. These features make Fe- and Cu-modified samples more promising for use in water flow treatment systems.

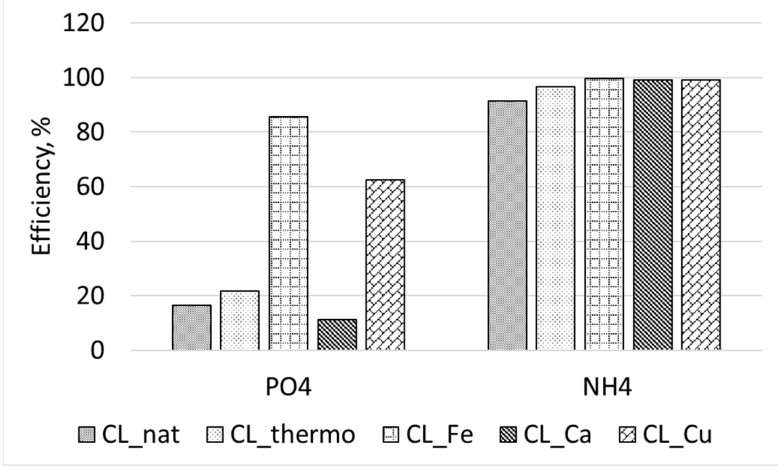

**Figure 7.** The efficiency of treatment.

## 4. Conclusions

Ammonia and phosphate removal from aqueous solutions using a novel metal and microwave-pretreated clinoptilolite was investigated. It was found that calcination at 550 °C leads to a decrease in the BET surface area in clinoptilolite, and the micropore area, unlike microwave irradiation. Pretreatment with $FeCl_3$ and $CuCl_2$ under microwave irradiation leads to an increase in the micropore area, although the BET surface area slightly decreases. The order for $S_{BET}$ values for clinoptilolite was CL-Fe > CL > CL-Cu > CL-Ca > CL-thermo. The shape of the curve representing the variations of $N_2$ amount adsorbed as a function of pressure corresponds to an IV-type isotherm typically characteristic of a mesoporous adsorbent.

Fe-modified and calcinated clinoptilolite showed the highest $NH_4^+$ adsorption capacity. Ca-modified samples demonstrated the lowest maximum adsorption capacity. The process of $PO_4^{3-}$ adsorption in natural and all modified samples is best described using the Langmuir—Freundlich model, confirming the assumption of phosphate physical adsorption on the heterogeneous surface of the clinoptilolite. The difference in microporosity and the specific surface area does not affect the $PO_4^{3-}$ sorption efficiency. Fe- and Cu-modified microwave-irradiated clinoptilolite has the best phosphate sorption characteristics. Fe-modified and calcinated clinoptilolite showed the highest $NH_4^+$ adsorption capacity of 4.375 and 2.879 mg/g. Ca-modified samples demonstrated the lowest maximum adsorption capacity of 0.875 mg $NH_4^+$/g. The metal-modified samples exhibit a significantly higher phosphate sorption capacity (from 800.62 for Cu sample to 813.14 mg/g for the Fe-modified sample) than natural (280.86 mg/g) or calcinated samples (713.568 mg/g). Clinoptilolite behaves as a sorbent for $PO_4^{3-}$ and as an ion exchanger for $NH_4^+$ ions.

Natural and calcinated clinoptilolite $NH_4^+$ adsorption breakthrough curves show more rapid saturation than all metal-modified samples. The treatment efficiency of Fe-, Cu-, and Ca-modified samples varies from 99.1 to 99.7%. The $PO_4^{3-}$ breakthrough curves of natural, calcinated, and Ca-modified samples reached the saturation point earlier and with lower efficiency than Fe- and Cu-clinoptilolite. These features make Fe- and Cu-modified samples more promising for use in water flow treatment systems.

**Author Contributions:** Conceptualization, K.S., I.F., A.M. and J.Š.; methodology, K.S., A.M. and J.Š.; validation, K.S., I.F. and J.M.; formal analysis, K.S., I.F. and J.Š.; investigation, K.S., I.F., A.M., J.Š. and J.M.; resources, K.S., I.F., A.M. and J.M.; data curation, K.S., A.M. and J.Š.; writing—original draft preparation, K.S., I.F., A.M., J.Š. and J.M.; writing—review and editing, K.S., A.M. and J.Š.; visualization, K.S., I.F., A.M. and J.M.; supervision, K.S. and A.M. All authors have read and agreed to the published version of the manuscript.

**Funding:** This research was funded by Research Council of Lithuania, according to the project "Sustainable technology of wastewater treatment by environmentally friendly modified natural sorbents for removal of nitrogen, phosphorus and surfactants", financing agreement No. S-LU-22-1 (EDINA code E22T040022).

**Data Availability Statement:** Data sharing not applicable.

**Conflicts of Interest:** The authors declare no conflict of interest.

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
