# Peer review of "Adsorption of Ammonium Ions and Phosphates on Natural and Modified Clinoptilolite: Isotherm and Breakthrough Curve Measurements"

_water, doi:10.3390/w15101933_

Round 1

Reviewer 1 Report (Previous Reviewer 1)

The authors have improved the quality of the paper and must be accepted for publication.

Author Response

Response to Reviewer 1 Comments

Point 1: The authors have improved the quality of the paper and must be accepted for publication.

Response 1: Thank you very much for your note.

Reviewer 2 Report (Previous Reviewer 2)

Title: ADSORPTION OF AMMONIUM IONS AND PHOSPHATES 2 ON NATURAL AND MODIFIED CLINOPTILOLITE: ISO- 3 THERM AND BREAKTHROUGH CURVE MEASUREMENTS

My comments are given here,

1. English grammar should be improved.

2. Mechanism of adsorption should be given in revised manuscript.

3. Comparison table with other absorbents used in literature should be given to highlight the importance of study.

4. Reusability of the adsorbent should also given in revised manuscript.

5. Cite this article in revised manuscript.

Rational synthesis and characterization of highly water stable MOF@ GO composite for efficient removal of mercury (Hg2+) from water

No

Author Response

Response to Reviewer 2 Comments

Point 1: English grammar should be improved.

Response 1: English grammar is revised.

Point 2: Mechanism of adsorption should be given in revised manuscript.

Response 2: The mechanism of adsorption is explained in lines 319-321 and 355-358: the sorption mechanisms proved the ion exchange in the case of ammonium and the formation of inner-sphere complexes with the functional groups M-OH (M: Al3+, Fe3+, Mn2+) in the case of phosphate. Hydrated metal oxides such as Fe(III) and Cu(II) have been more extensively studied for phosphate removal because of their strong ligand sorption (HPO4 2− and H2PO4 ), through the formation of inner-sphere complexes or through outer-sphere complexes.

Point 3: Comparison table with other absorbents used in literature should be given to highlight the importance of study.

Response 3: Thank you very much for your note. A comparison with other sorbents is given in lines 343-351. According to the authors, there are few similar studies in which ammonium ions and phosphate ions are removed from aqueous solutions at the same time. The conditions are very different, so it is not appropriate to present the table.

Point 4: Reusability of the adsorbent should also given in revised manuscript.

Response 4: Thank you very much for your note. Reuse of sorbents was not studied in the article. The authors consider the possibility of using sorbents that have absorbed nitrogen and phosphorus as plant fertilizers. It is noted in lines 351-355.

Point 5: Cite this article in revised manuscript. “Rational synthesis and characterization of highly water stable MOF@ GO composite for efficient removal of mercury (Hg2+) from water’’.

Response 5: Thank you very much for your note. This new and important article is now cited in our article introduction.

Reviewer 3 Report (Previous Reviewer 4)

The comments I made in the review have been corrected. The article can be accepted for publication

Author Response

Response to Reviewer 3 Comments

Point 1: The comments I made in the review have been corrected. The article can be accepted for publication.

Response 1: Thank you very much for your note.

This manuscript is a resubmission of an earlier submission. The following is a list of the peer review reports and author responses from that submission.

Round 1

Reviewer 1 Report

1. The findings of the work must be given in the abstract.

2.  Add more techniques to remove pollutants using the following reference https://doi.org/10.1002/9781119818915.ch11.

3. The novelty of the work should be properly addressed in the introduction.

4. In heading 3. R is missing.

5. Why correlation coefficient (R2) was not calculated?

Reviewer 2 Report

Paper Title: ADSORPTION OF AMMONIUM IONS AND PHOSPHATES ON NATURAL AND MODIFIED CLINOPTILOLITE: ISOTHERM AND BREAKTHROUGH CURVE MEASUREMENTS.
Manuscript Number: 2329426
Article Type: Research Article
Journal Name: Water
Reviewers Comments;
This paper entitled "
ADSORPTION OF AMMONIUM IONS AND PHOSPHATES ON NATURAL AND MODIFIED CLINOPTILOLITE: ISOTHERM AND BREAKTHROUGH CURVE MEASUREMENTS". Authors have written manuscript which is scientifically and well-organized so, I do recommend this paper for publication after major revision. However following comments should be addressed. These comments are given below:

1.       There are some grammar mistakes these should be removed from manuscript.

2.       In abstract first two lines should be removed which are related to literature. In abstract you write your work and its findings. Abstract start from This research or in this study.

3.       In abstract it is good for reader if you add concise findings concluded from spectroscopic techniques like BET.

4.       It is good to perform SEM, EDS, FTIR and PXRD of the synthesized compounds to confirm their structure and morphology.

5.       It is good to add which model (Langmuir, Freundlich etc.) is best and its coefficient value should be given in abstract.

6.       In abstract also give BET surface area.

7.       In abstract also give maximum adsorption capacities shown by adsorbents.

8.       It is good to perform adsorption parameters like time effect, temperature effect, Concentration of pollutants and adsorbents.

9.       In abstract you have given “in dynamic conditions revealed high NH 4+ and sufficient PO 4 3- ions capture on modified clinoptilolite. Fe- and Cu-modified samples were suggested as more promising for use in water flow purification systems.” Fe and Cu shows good results why?

10.   Also mention in intro their WHO recommended values for adsorbates used in this study.

11.   In intro you have given “Wastewater from private households” what do you means by private households.

12.   Study gap is not given in this study at the end of intro.

13.   In materials method you have used chemicals. These metal salts were anhydrous or hydrated? Also mention here from which company they were received.

14.   In BET surface area oC is given in different ways. Give in same ways. Give space or not but do same.

15.   Only BET is not sufficient for adsorbents also perform FTIR, EDX before and after adsorption process to confirm adsorption.

16.   In adsorption also mention concentrations of removed adsorbates.

17.   Also mention their chemical names to prepared their solutions.

18.   Also mention equilibrium time in conclusions.

19.   Also give a comparison table with other reported adsorbents for efficiency of your study.

20.   Give references for all equations used in study.

21.   In results and discussions R is not written before results.

22.   Reusability of adsorbents is also very important. Do you have performed reusability?

23.   You have synthesized different adsorbents for water purification. Do you have performed any test to check their stability in water. How much they are stable in water, because Cu metal in high concentrations also cause water pollution.

24.   These different articles should be very helpful for this study. Also cite these articles in revised manuscript.

A comprehensive review on technological advances of adsorption for removing nitrate and phosphate from waste water

Engineering of Zirconium based metal-organic frameworks (Zr-MOFs) as efficient adsorbents

Nanoscale materials as sorbents for nitrate and phosphate removal from water

Design, Synthesis and Spectroscopic Characterizations of Medicinal Hydrazide Derivatives and Metal Complexes of Malonic Ester,

Fabrication of a novel bifunctional nanocomposite with improved selectivity for simultaneous nitrate and phosphate removal from water

Metal-Organic Frameworks Derived Electrocatalysts for Oxygen and Carbon Dioxide Reduction Reaction,

A comprehensive review on technological advances of adsorption for removing nitrate and phosphate from waste water

Rational synthesis and characterization of highly water stable MOF@ GO composite for efficient removal of mercury (Hg2+) from water,

Nitrate and phosphate removal by chitosan immobilized Scenedesmus

Effect of metal atom in zeolitic imidazolate frameworks (ZIF-8 & 67) for removal of Pb2+ & Hg.

Zeolite supported Fe/Ni bimetallic nanoparticles for simultaneous removal of nitrate and phosphate: synergistic effect and mechanism

Synthesis of New Series of Phenyldiazene Based Metal Complexes for Designing Most Active Antibacterial and Antifungal Agents.

Batch-wise nitrate removal from water on a surfactant-modified zeolite

Reviewer 3 Report

This manuscript investigates ammonia and phosphate removal from aqueous solutions by using a metal and microwave-pretreated clinoptilolite.

1)     There are some typo and grammatical errors in the manuscript.

2)     Clinoptilolite has already been used in many studies. What is the novelty of this manuscript? It should be described.

3)     What is the research hypothesis?

4)     Materials should be discussed in the materials and methods section. Also, modification method should be given.

5)     Ca-modified samples demonstrated the lowest maximum adsorption capacity. Why?

6)     Authors claim that “These features make Fe- and Cu-modified samples more promising for use in water flow purification systems.” How? Why?
